# Microtubule and Actin Cytoskeletal Dynamics in Male Meiotic Cells of *Drosophila melanogaster*

**DOI:** 10.3390/cells11040695

**Published:** 2022-02-16

**Authors:** Anna Frappaolo, Roberto Piergentili, Maria Grazia Giansanti

**Affiliations:** Istituto di Biologia e Patologia Molecolari del CNR, c/o Dipartimento di Biologia e Biotecnologie, Sapienza Università di Roma, Piazzale A. Moro 5, 00185 Roma, Italy; anna.frappaolo@cnr.it (A.F.); roberto.piergentili@uniroma1.it (R.P.)

**Keywords:** spindle microtubules, cytoskeleton, male meiosis, cytokinesis

## Abstract

*Drosophila* dividing spermatocytes offer a highly suitable cell system in which to investigate the coordinated reorganization of microtubule and actin cytoskeleton systems during cell division of animal cells. Like male germ cells of mammals, *Drosophila* spermatogonia and spermatocytes undergo cleavage furrow ingression during cytokinesis, but abscission does not take place. Thus, clusters of primary and secondary spermatocytes undergo meiotic divisions in synchrony, resulting in cysts of 32 secondary spermatocytes and then 64 spermatids connected by specialized structures called ring canals. The meiotic spindles in *Drosophila* males are substantially larger than the spindles of mammalian somatic cells and exhibit prominent central spindles and contractile rings during cytokinesis. These characteristics make male meiotic cells particularly amenable to immunofluorescence and live imaging analysis of the spindle microtubules and the actomyosin apparatus during meiotic divisions. Moreover, because the spindle assembly checkpoint is not robust in spermatocytes, *Drosophila* male meiosis allows investigating of whether gene products required for chromosome segregation play additional roles during cytokinesis. Here, we will review how the research studies on *Drosophila* male meiotic cells have contributed to our knowledge of the conserved molecular pathways that regulate spindle microtubules and cytokinesis with important implications for the comprehension of cancer and other diseases.

## 1. An Overview of Spermatogenesis in *Drosophila melanogaster*

Spermatogenesis is the complex differentiation process that leads to the formation of mature sperms from diploid germ stem cells within the testis. *Drosophila melanogaster* spermatogenesis offers a powerful cell system in which to investigate cytoskeleton organization and remodeling during the different events that characterize germ cell differentiation, including germ cell development, meiosis, and cytokinesis. Each *Drosophila* adult testis is a long blind-ended tube, in which the different stages of spermatogenesis are arranged in a spatiotemporal order, with the spermatogonia and spermatocyte divisions residing in the first third of the testis [1]. An overview of the sequential events described below is summarized in Figure 1.

Spermatogenesis starts at the apical tip of the testis, which hosts two stem cell populations, namely the germline stem cells (GSCs) and somatic cyst stem cells (CySCs) [2]. A group of 8–10 GSCs divides asymmetrically to generate another GSC and a daughter cell that leaves the niche and differentiates into a gonialblast [3]. Each gonialblast undergoes four transit-amplifying mitotic divisions. Cytokinesis is incomplete in each division; male germ cells undergo cleavage furrow ingression but do not complete abscission, leaving stable intercellular bridges of 1–2 μm in diameter called ring canals (RCs) that interconnect the resultant 16 spermatogonia (SGs) [4,5]. Besides the presence of RCs, the male germ cells within each cyst are connected by the fusome, a branched membrane and cytoskeletal-rich organelle that passes through each RC and coordinates intracyst signaling [6,7,8,9]. The 16 SGs in each cyst progress to the premeiotic S phase and then embark on the G2 phase, switching to the growth and meiosis-specific transcription programs [5,10]. During the extended G2 phase, which lasts 90 h, the cell volume grows 25-times, while spermatocytes undergo robust gene expression [1,11,12]. Mature spermatocytes undergo two meiotic divisions in rapid succession, with incomplete cytokinesis, resulting in 64 haploid round spermatids connected by RCs [1,13,14]. Following meiosis, all the mitochondria, contained in each spermatid, aggregate around the basal body at one side of the nucleus and fuse to form a complex interlaced structure named the nebenkern [15]. During spermiogenesis, spermatids undergo a differentiation phase, during which their nuclei dramatically shrink and most of their cytoplasm and organelles are lost. The nebenkern elongates and splits into two symmetrical halves surrounding the growing axoneme, while the acrosome, a membrane bound organelle required for fertilization, forms from the acroblast (a Golgi derived organelle). The final step of spermiogenesis is the individualization/coiling process: starting from the acrosome, membranous cones expand towards the distal end of the elongating sperm tail; then sperm tails coil and transfer into the seminal vesicles [15]. Compelling reviews on *Drosophila* spermatogenesis in its entirety are present in literature [1,16] and go beyond the scope of the present work. Here, we will focus on the cytoskeleton changes that characterize male meiosis and cytokinesis.

## 2. Translational Control and Cytoskeletal Remodeling in *Drosophila* Spermatocytes

In *Drosophila* testes, cytoskeleton remodeling at the G2/M transition of the male meiotic division is under the control of the cell cycle regulating phosphatase twine/Cdc25, which removes inhibitory phosphorylation from Cyclin B/Cdk1 [17,18,19,20]. Testes from mutants in *twine*/*Cdc25* display spermatid differentiation, despite a failure to undergo meiotic divisions, indicating that the activation of spermatid differentiation is not dependent on prior completion of meiosis [21,22]. Translational control has a crucial role in regulating the G2/M transition in male meiotic cells. Although the transcript for core cell cycle protein Cyclin B1 (CycB) is expressed in growing spermatocytes, stage-specific repression of translation restricts CycB protein accumulation in the spermatocyte cytoplasm to the late G2 phase (Figure 1) just before chromatin condensation [22,23,24,25].

A key step for translational control is the recognition of the mRNA 5′ cap structure by the eukaryotic binding complex eIF4F [26]. This complex, which consists of the cap-binding protein eIF4E, the RNA helicase eIF4A and the scaffolding protein eIF4G, binds the poly(A)-binding protein to promote circularization of mRNAs and ribosome 40S subunit recruitment [27]. *Drosophila melanogaster* encodes multiple isoforms of both eIF4E and eIF4G. Two research groups identified eIF4-G2 as a novel orthologue of eukaryotic initiation factor 4G (eIF4G), required for translational control during male germ cell differentiation [23,25]. Baker and Fuller showed that germ cells from *eIF4G2* mutant testes skip the major events of meiotic division but initiate spermatid differentiation, resulting in aberrant spermatids with large nuclei. Consistent with the meiotic defects, *eIF4G2* mutants exhibit defects in translation of the core cell cycle regulatory proteins Twine and CycB in mature spermatocytes [23]. Franklin-Dumont and coworkers showed that, during the meiotic G2 phase, spermatocytes from *eIF4G2* mutant males do not reach their correct size and fail to undergo meiotic division and differentiate properly. Based on these results they suggest that a checkpoint stops male meiosis and cell differentiation when the spermatocytes have not accumulated a sufficient cell mass [25]. The testis-specific translation factors eIF4E-1 and eIF4E-3 are also required for translational control. Testes from males carrying mutations in the eIF4E-1 and eIF4E-3 translation factors display both defects in chromosome condensation and segregation and impaired cytokinesis [28,29].

The *Drosophila* C2H2-zinc finger protein Doublefault (Dbf), primarily expressed in the testes, is also involved in controlling mRNA translation and CycB expression in male meiotic cells [30]. Dbf forms a complex with the RNA binding protein Syncrip/hnRNPQ [31,32], and the interaction is dependent on RNA targets [30]. Importantly, Syncrip is the ortholog of mammalian SYNaptotagmin-binding Cytoplasmic RNA-Interacting Protein (SYNCRIP/hnRNPQ), which is localized in the RNA transport granules associated with dendritic mRNAs of hippocampal neurons [33,34,35]. *Drosophila* Syncrip protein was also shown to regulate the localization and translation of specific mRNAs in oocytes and neuromuscular junctions [31,32]. Dbf protein binds *cycB* mRNA and is required for normal accumulation of CycB protein, prior to the first meiotic division. [30] Despite the defective CycB expression, *dbf* mutant spermatocytes embark on highly irregular meiotic division. Loss of Dbf affects multiple aspects of meiotic division, including centriole disengagement, centrosome structure, chromosome segregation and cytokinesis. In *Drosophila* germ cells, gene transcription shuts down after the meiotic G2/M boundary, and most of the gene products required for meiosis and spermiogenesis are transcribed in G2 spermatocytes [1,36,37,38]. Thus, the pleiotropic phenotype of *dbf* mutant males can be correlated with the requirement for *dbf* for the translation of specific mRNAs in spermatocytes [30]. Dbf protein is also required to regulate centriole and axoneme structures. Loss of Dbf results in an increased length of centrioles and cilia-like projections in dividing spermatocytes. These phenotypic defects can be associated with the requirement for Dbf for the expression of specific molecular components of the centriole/cilium [30]. However, Dbf protein localization at the polar regions of meiotic spindles suggests that it might locally facilitate the translation of ciliary proteins and ciliogenesis. Consistent with this hypothesis, ciliary basal bodies and centrosomes contain RNA-processing proteins and share molecular components with stress granules and P-bodies [39,40,41,42]. Moreover, several translation factors have been localized to the centrosomes [43], leading to the suggestion that centrosomal translation would facilitate the translation of ciliary proteins during ciliogenesis.

Importantly, Fingerhut and Yamashita recently identified novel cytoplasmic ribonucleoprotein (RNP) granules in late spermatocytes that contain mRNAs for the testis-specific axonemal dynein heavy chains, along with the AAA+ (ATPases associated with diverse cellular activities) proteins Reptin and Pontin [44]. They further showed that these RNPs granules segregate during the meiotic divisions and facilitate the accumulation and efficient incorporation of axonemal dynein proteins into the axoneme during spermiogenesis [44]. Because *dbf* mutants also display a defective assembly of the canonical 9+2 axoneme cartwheel in elongating spermatids, it should be important to investigate how Dbf protein contributes to flagellar axoneme elongation and whether it coordinates axonemal dynein translation together with Reptin and Pontin.

## 3. Centrioles and Centrosomes in Male Meiotic Cells

The primary spermatocytes of *Drosophila melanogaster* contain two centrosomes that act as the major microtubule nucleation sites for meiotic spindle assembly [45].

Centrosomes consist of a pair of nine-fold symmetrical centrioles embedded in a proteinaceous matrix, known as the pericentriolar material (PCM) [45,46,47]. In contrast with vertebrate centrioles, there are no distal and subdistal appendages on the mother centriole in *Drosophila* [48]. Centrioles also play a distinct function outside the centrosome as basal bodies, serving as templates for cilia and flagella [45,48,49]. Several recent reviews have been focused on centriole duplication and biogenesis and on the structure and functions of the *Drosophila* centrosome [45,50]. The centrioles of mature spermatocytes are by far longer than those in SGs and somatic cells. Indeed, centrioles undergo a dramatic elongation during the growth phase of G2 spermatocytes, reaching a size of 2.3 µm at the metaphase of meiosis I [51]. The core components required for centriole assembly and duplication were initially identified during genetic screens in *Caenorhabditis elegans* and have counterparts in *Drosophila* [52,53]. An analysis of *Drosophila* centrosomes by super-resolution light microscopy provided a valuable method to analyze centrosome architecture and to define the reciprocal localization of centriolar and PCM proteins [54,55,56,57]. Sas-6 and Ana2 proteins (the orthologs of human Sas-6 and STIL) concentrate to the innermost core of centrioles [57,58]. Sas-4/CPAP mediates the connection between the centriole cartwheel and the microtubule wall and interacts with the α/β tubulin dimers, Ana2/STIL and Cep135 [59,60,61,62,63]. In *Drosophila* S2 cultured cells, centriole duplication occurs prior to G1, after the disengagement of the mother and daughter pair of centrioles at the end of mitosis [56]. Polo (PLK1) kinase regulates the disengagement of mitotic centrioles in human cells [64,65]. In agreement with findings in human cells, pharmacological inhibition or spermatocyte-specific depletion of Polo kinase in *Drosophila* testes impairs centriole separation that occurs during anaphase I [30,66,67]. A phenotypic analysis in *Drosophila myt1* mutants uncovered a role for the kinase Myt1 in centriole disengagement in male meiotic cells [67]. The inhibition of Cdk1/CyclinA by Myt1 kinase is required to maintain centriole engagement during the premeiotic G2 phase of *Drosophila* male meiosis. Vadarajan and coworkers showed that spermatocytes from *myt1* mutant males display premature centriole disengagement during G2-phase arrest and form multipolar spindles during meiosis. Spermatocyte-specific depletion of Polo kinase suppressed the centriole phenotypic defects associated with *myt1* loss of function, indicating that the mechanisms of Polo-mediated centriole disengagement depend on Myt1 regulation of CyclinA/Cdk1.

The conserved Ser/Thr kinase Polo-like kinase 4 (Plk4) controls the events that lead to centriole assembly and duplication in both *Drosophila* and humans [68,69,70]. The loss of Plk4, by either mutation or RNAi, impairs centriole replication and basal body formation, whereas its overexpression results in centriole amplification [70]. Asterless (Asl), the ortholog of mammalian CEP152, plays an essential role in recruiting Plk4 to the centrosome [71]. The amino terminus of Asl interacts with the cryptic Polo box of Plk4, whereas its carboxy terminus binds the centriolar protein Sas-4 (CPAP in humans) to help build the PCM [71]. Galletta and coauthors performed a centrosome interactome study applying a yeast-two-hybrid screen and used this analysis to gain further insight into the protein organization within the centrosome [72]. Their study led to the identification of several molecular partners of Plk4, namely Ana1, Ana2, Asl, CP110, Bld10 and Plk4 itself [72]. They further showed that Plk4 phosphorylates Bld10, which is the ortholog of human CEP135 [73], and that this phosphorylation controls its interaction with Asterless (Asl) and the radial positioning of Asl on centrioles. Plk4 also phosphorylates the core centriole component Ana2, which, in turn, recruits Sas-6 to the mother centriole, leading to cartwheel assembly [56,74,75,76]. Many studies have demonstrated that the nine-fold symmetry of the cartwheel and the centriole results from the arrangement of the nine interacting dimers of Sas-6 protein [58,77,78,79].

## 4. Centrosome Pericentriolar Material Assembly (PCM) in Male Meiotic Cells

At the G2/M transition, centrosomes undergo maturation, resulting in PCM expansion and the recruitment of γ-tubulin, the universal microtubule nucleating component from the centrosome. Studies in male germ cells and other cell types have contributed to identify the proteins that regulate this process in *Drosophila* [45]. γ-tubulin does not act as a monomer but in a complex with other proteins. The simplest one, characterized in *Drosophila* by Oegema and coauthors, named γ-tubulin small complex (γ-TuSC), is a tetrameric complex which contains γ-tubulin dimers and Dgrip84 and Dgrip91, the counterparts of the yeast Spc98p and Spc97p proteins, in a 2:1:1 stoichiometry [80]. The larger *Drosophila* γ-tubulin complex (γTuRC) that forms the foundation for nucleating microtubules contains several γ-TuSC and at least four other proteins [81,82]. Polo/Polo-like kinase 1 (Plk-1) plays an essential role in centrosome maturation, as shown by the original observations that *polo* mutants displayed defective spindle poles and an altered localization of the centrosomal protein CP190 [83]. In human and *Drosophila* cells, the concentration of γ-tubulin, Spd-2/Cep192, Centrosomin (Cnn)/CDK5RAP2 and Pericentrin-like-protein (Plp)/Pericentrin increases during centrosome maturation [54,84,85]. Using a combination of live imaging, Electron Microscopy and Electron Tomography, Roque and coauthors examined the role of *Drosophila* Plp in the centrosomes of two different cell types, namely the Sensory Organ Precursor lineage of the pupal notum and the spermatocytes [86]. In both cell types, *Plp* mutant centrioles undergo premature disengagement (and thus over-duplicate) and organize fewer microtubules during interphase. In worms, SPD2 function is required for recruiting the ZYG-1 polo kinase (the ortholog of the fly Plk4) to the mother centriole [52,53]. Conversely, the function of *Drosophila* Spd-2 is dispensable for centriole duplication but is essential for the recruitment of PCM in somatic cells and in spermatocytes, where it also maintains cohesion between the two centrioles [87,88]. Many research studies have been focused on the role of Cnn in the assembly of mitotic PCM [89,90,91,92,93]. Cnn recruitment around mother centrioles depends on Spd-2 function, and a small central region of Cnn, termed phosphoregulated multimerization (PReM) domain, contains several phosphorylation sites, including those for Polo kinase [94,95]. Cnn and Spd-2 are incorporated into the PCM close to the centrioles and then spread outward to form a platform that recruits other PCM proteins [94,95]. When either Spd-2 or Cnn is absent, PCM assembly is greatly reduced, whereas it is abolished by the loss of both proteins [95]. Conduit and coauthors showed that Asl controls the recruitment of DSpd-2 and Cnn exclusively to the mother centrioles during the mitotic PCM assembly of *Drosophila* embryos [95]. The centriolar proteins Asl and Sas-4 have also been implicated in PCM assembly in dividing spermatocytes [71,96,97,98,99]. However, further studies will be needed to dissect the role of centriolar proteins in the PCM assembly in male meiotic cells.

## 5. The Centrosome Is Essential to Assemble a Functional Bipolar Spindle in Spermatocytes

Accurate chromosome segregation during male mitotic and meiotic divisions relies on the assembly of a microtubule-based bipolar spindle [46].

During mitotic divisions of *Drosophila* male GSCs, centrosome function is required to direct mitotic spindle orientation in the niche. The stereotypical orientation of centrosomes within stem cells plays an important role to orient mitotic spindles perpendicular to the hub–stem cell interface, resulting in one daughter, which inherits the attachment to the hub and maintains stem cell identity, whereas the other one is displaced away and commits to differentiation [100,101,102]. Moreover, a mechanism, known as the centrosome orientation checkpoint, exists, which requires wild type function of Cnn and E cadherin and regulates cell-cycle progression, inducing cell cycle arrest in response to centrosome misorientation [103,104]. Once this checkpoint is cleared, the gonialblast undergoes four transit-amplifying, symmetric mitotic divisions.

Centrosome-, chromatin- and microtubule-mediated mechanisms coexist and differently contribute to bipolar spindle formation in different cell types [46,105]. Although dispensable during mitotic divisions in GSCs and SGs, centrosomes are the major microtubule organizing centers in spermatocytes and are strictly required for normal bipolar spindle formation and chromosome segregation in these cells [105]. A major advantage in examining the phenotypes of mutants affecting spindle assembly during male meiotic divisions is that the spindle assembly checkpoint is not robust in spermatocytes, resulting in a small delay in meiotic progression [106,107]. The *Drosophila melanogaster* genome encodes two gamma tubulin proteins, and only γ-tubulin23C is expressed in the testes [108,109,110]. Spermatocytes carrying mutations in either *γ-tubulin23C* or *dd4* (the *Drosophila* gene encoding Dgrip91) initially assemble large arrays of microtubules but fail to form or maintain bipolar spindles [111,112]. A cytological analysis of male meiosis in mutants in different centriole/PCM components, such as *asl*, *cnn* and *spd-2,* indicated that centrosomes are essential for the assembly of a functional bipolar spindle in spermatocytes [88,91,96]. Consistent with the prominent role of centrosomes in driving bipolar spindle formation, secondary spermatocytes from *solofuso* and *fusolo* mutants are able to nucleate astral microtubules and assemble bipolar spindles in the absence of chromosomes [113]. During the prolonged G2 phase growth of spermatocytes, the centrioles dissociate from the nuclear envelope and dock at the plasma membrane, where they assemble short cilia on all four centrioles [114]. At the onset of meiosis, the centriole pairs (together with their short cilia) dislodge from the plasma membrane, move back towards the nucleus and nucleate astral microtubules. Rebollo and coauthors analyzed the effects of two experimental conditions in which centrosomes remain anchored to the plasma membrane, throughout meiosis I of *Drosophila* male meiosis [115]. By using time-lapse confocal microscopy in these spermatocytes, they were able to report centrosome-independent microtubule nucleation and bipolar spindle assembly in correspondence with the nuclear region, away from the asters. The centrosome-independent microtubule nucleation was shifted in time with respect to the nuclear envelope breakdown and occurred in foci in close proximity with the membranes that enclose the nuclear region. These observations led them to suggest that the fenestrated nuclear envelope plays an important role in microtubule nucleation in *Drosophila* spermatocytes and that both centrosomal and noncentrosomal microtubules contribute to the robustness of bipolar spindle assembly.

Recent studies have revealed the existence of a novel acentrosomal microtubule-mediated nucleation pathway, which involves microtubules within the spindle and the conserved hetero-octameric protein complex Augmin [46,116,117,118,119,120,121,122]. In *Drosophila* S2 cells and embryos, Augmin regulates the increase of the microtubule density within mitotic spindles to stabilize kinetochore fibers, chromosome alignment and spindle bipolarity [116,118,122,123,124,125]. In human cells HAUS, the human Augmin complex, is required for mitotic assembly and centrosome integrity [123]. It is amply accepted that Augmin targets γ-tubulin complexes along the spindle microtubules and promotes microtubule nucleation [46]. Savoian and Glover recently studied the molecular mechanisms that control acentrosomal spindle assembly in *Drosophila* spermatocytes by using both time lapse fluorescence microscopy and immunofluorescence analysis [126]. They demonstrated that γ-tubulin and the Augmin targeting complex are required for acentrosomal microtubule nucleation at kinetochores or near the chromosomes. In contrast with fly somatic cells, Augmin is not recruited on male meiotic spindles and accumulates to the kinetochores. Moreover, the process is regulated by the inhibitory effects of Polo kinase on Augmin, which controls the spindle microtubule density and architecture.

## 6. Spindle Scaling in Primary Spermatocytes

‘Spindle scaling’ is the characteristic of mitotic and meiotic spindles to adjust their size according to the volume of the hosting cell, at least in some organisms and in specific stages of their development. Spindle size in mitotically dividing cells is largely variable, with a pole-to-pole distance spanning from 10.2 μm ± 2.9 in *C. elegans* to 53.5 μm ± 5.9 in *X. laevis* [127], i.e., a difference of eight-fold. The question of spindle size is not trivial, because it has been shown that a consistent cell diameter to spindle length ratio is maintained during early development across different organisms [127]. The molecular mechanisms underlying spindle scaling are largely unknown, although research data suggest a correlation between the spindle size and the relative abundance of its components [128]. Although meiotic spindles do not scale as closely to the egg size in females, with only a few exceptions recorded to date [127], the male meiosis of Drosophilids provides an amenable cell system to study the mechanisms underlying the spindle architecture and its size regulation. In contrast with mammalian cells that show comparable sizes of male meiotic and mitotic spindles [129], primary spermatocyte nuclei and spindles in flies show ample dimensional variation within different *Drosophila* species [130]. Moreover, male meiotic spindles are substantially larger than the spindles of somatic cells and those of mammals. The sperm tail length in Drosophilids is extremely variable as well, ranging from that of *D. persimilis* (0.32 mm) to the giant ~60 mm sperm produced by *D. bifurca* [131,132], a whopping ~185x ratio. Several studies aimed to understand the evolutionary strategy behind this difference in size, with the notion that longer sperm tails are inversely correlated with sperm numbers [133] and directly with the timing of male maturity [132]. Far less investigated is the relationship between spindle size and sperm tail length. It has been noted that *D. persimilis* shows smaller primary spermatocyte nuclei and shorter tails, while *D. bifurca* shows opposite features; the analysis of four additional species of intermediate size for both parameters revealed a linear, positive correlation between these two variables [130]. The same authors reported a positive correlation between spindle length and area [130]. These data suggest that due to the extraordinary length of the sperm tail in these species compared to mammals (the average length of the *H. sapiens* sperm being 50–60 µm), primary spermatocytes need to store large amounts of tubulin that will be used for building the axoneme during spermiogenesis. Research studies in *X. laevis* embryos reported a correlation between the cytoplasm volume and the spindle size [134,135], as well as between the cell size and the cell cycle speed, with smaller cells dividing faster than larger ones [136,137]. A similar correlation between the cell size and the spindle size has been described in other species, such as *D. rerio* [138], *C. elegans* [139,140] and *P. lividus* [139]. We also recall here that during the growth phase and just before the meiotic spindle assembly of *Drosophila* males, the β1-tubulin isotype is replaced by the testis-specific β2 isoform, and most of this isoform is enriched in the sperm tails during the elongation process [141,142]. In turn, the accumulation of the molecular components of such long axonemes is reflected in the large size of dividing spermatocytes and their spindles that store the large pools of tubulin. Also notable is the microtubule-dependent inheritance of mitochondria during male meiotic divisions, which are also located along the mature sperm tail and provide the energy to move such long flagella. During each meiotic division, mitochondria along the central spindle microtubules occur, which allows the equal partitioning of these organelles between the two daughter cells at the end of cytokinesis (see below) [1,143]. On the other hand, mitochondria serve as microtubule-organizing centers, providing a structural platform for microtubule remodeling during sperm tail elongation [144]. Taken together, these data indicate that mitochondria are also involved in this highly controlled process.

## 7. The Advantages Offered by *Drosophila* Male Meiotic Cells to Explore Cytokinesis

Cytokinesis, the final event of each mitotic and meiotic division, partitions the cytoplasm and the duplicated genome of the mother cell into two daughter cells [145]. In *Drosophila* dividing spermatocytes, as in most animal cells, cytokinesis is accomplished by the contraction of the contractile ring, a structure composed of F-actin filaments and non-muscle Myosin II, assembled on the inner face of the plasma membrane around the cell equator [13,14,146]. This intricate process involves the coordinated reorganization of the spindle microtubules and the actin cytoskeleton and plasma membrane remodeling at the cleavage furrow. *Drosophila* spermatogenesis offers several advantages for investigating the molecular mechanisms that regulate cytokinesis. As already mentioned, *Drosophila* spermatocytes are larger than most somatic cells and display comparable large microtubule meiotic spindles and clear actomyosin rings (Figure 2), making these cells highly suitable for immunofluorescence and in vivo analysis of the cytokinetic structures.

Moreover, since cytokinesis is incomplete in male germ cells, clusters of primary and secondary spermatocytes undergo meiotic division in synchrony in each cyst, providing a large number of dividing cells for quantitation of phenotypic defects. These cytological characteristics, together with the availability of sophisticated genetic tools in flies, have contributed to the success of male meiotic cells as a model system for genetic and molecular analysis of cytokinesis. Evidence indicates that although male meiotic cytokinesis lacks abscission, the molecular machinery required for cleavage furrow formation and ingression is largely conserved when compared with other animal dividing cells [14]. During male meiotic divisions of *Drosophila*, the nuclear envelope does not disassemble by metaphase I but becomes fenestrated near the spindle poles [51,147]. A small number of centrosome-nucleated microtubules penetrate beyond the fenestration to contact kinetochores, whereas the majority of metaphase spindle microtubules remain outside the nuclear envelope [51,147]. From metaphase to telophase, the spindle-shaped nuclei are also surrounded by parafusorial membranes, a system of five to seven double membranes that derive from the endoplasmic reticulum (ER). During each meiotic division, mitochondria line up along the spindle microtubules outside of the parafusorial membranes, which ensures the equal partitioning of these organelles between the two daughter cells upon the completion of cytokinesis. Following the second meiotic division, all the mitochondria in each early round spermatid aggregate to one side of the nucleus and fuse to form a complex interlaced mitochondrial derivative called the nebenkern [1]. When examined by phase-contrast optics, cysts of wild-type early round spermatids contain a total of 64 interconnected spermatids, each containing a single phase-dark nebenkern associated with a single phase-light nucleus of a similar size [1]. Importantly, all the 64 spermatids within a wild-type cyst display nuclei and nebenkerne of identical sizes (Figure 1). This pattern allows the easy scoring of mutants that are defective in meiotic cytokinesis. Cytokinesis failures disrupt the partitioning of mitochondria into daughter cells during one or both meiotic divisions, resulting in spermatids containing two or four nuclei associated with an enlarged nebenkern. When defects in male meiotic cytokinesis are coupled with errors in chromosome segregation, the resultant spermatids contain a large nebenkern associated with multiple nuclei of variable size. In this context, it is important to recall that due to the lack of a stringent spindle assembly checkpoint in spermatocytes [106,107], *Drosophila* male meiosis allows one to investigate whether the gene products required for spindle formation and chromosome segregation play additional roles in later stages of cell division during cytokinesis [14,146].

## 8. The Role of Spindle Microtubules in Cleavage Furrow Formation in *Drosophila* Spermatocytes

In animal cells that undergo symmetric division, the cleavage site is located at the equatorial cortex, in a position that bisects the axis of chromosome segregation [145]. Anaphase spindle microtubules exert a key role in directing the position of the cleavage site during cytokinesis, as shown for the first time by spindle manipulation experiments in sand dollars embryos [148]. The mitotic spindle undergoes a profound reorganization after the anaphase onset, leading to the formation of the central spindle (CS), an array of antiparallel interdigitating microtubule bundles between segregated chromosomes [149]. However, after decades of studies, it is still controversial how the anaphase spindle delivers the signals that trigger cytokinesis and what part of the spindle (either the aster or the central spindle) is involved [150]. Cytological analysis of male meiotic divisions in mutants that are defective in cytokinesis has revealed the requirement for CS microtubules for the assembly of the F-actin rings and for the maintenance of these structures during cleavage furrow ingression [7,151,152,153]. Moreover, the phenotypes of mutants defective in spermatocyte cytokinesis have suggested the existence of a mutually dependent interaction between the central spindle microtubules and elements of the F-actin cytoskeleton. Spermatocytes from males carrying mutations in the genes required for the assembly of the CS structure, such as *klp3A*, display a primary defect in the spindle midzone formation, which determines a secondary defect in the F-actin ring assembly [7,153]. On the other hand, an identical phenotype characterized by defective central spindles and a failure to assemble a contractile ring is observed in dividing spermatocytes from males carrying mutations in the actin regulators profilin and formin or in the non-muscle-Myosin II regulatory light chain [7].

By using a time-lapse analysis of spermatocytes expressing β-tubulin-EGFP, Inoue and coworkers provided the first in vivo characterization of central spindle assembly and dynamics during male meiotic divisions [154]. Their analysis showed that the central spindle is made up of two different microtubule populations. The peripheral astral microtubules, which are important for cleavage site specification, contact the cortex during anaphase and bundle together to promote cleavage furrow formation. A distinct set of microtubules, named interior microtubules, resides within the highly fenestrated nuclear envelope that does not break down during meiotic divisions (see above). When cleavage furrow ingression initiates, the two populations of peripheral and interior microtubules coalesce, leading to the consolidation of the central spindle structure. Hypomorphic mutations in the gene *orbit*, which encodes a microtubule-associated protein, do not affect the initial steps of cytokinesis promoted by the bundles of peripheral astral microtubules but impair interior central spindle microtubules, leading to unstable cleavage furrow ingression. In agreement with these phenotypic defects, the wild-type Orbit protein in male meiotic cells concentrates in the region described as the spindle matrix [154,155].

Besides Orbit, other microtubule-associated proteins required for CS morphogenesis in *Drosophila* include microtubule bundling proteins and motor and signaling proteins [13,145]. The microtubule bundling protein Fascetto (Feo) protein, the ortholog of the highly conserved Protein Regulating Cytokinesis 1 (PRC1), is one of the cytokinesis markers in dividing *Drosophila* spermatocytes and concentrates in the central overlap region of the anaphase central spindles [156,157]. Although the cytokinesis phenotype associated with the loss of Feo has not been investigated yet, it has been shown that Feo is required for central spindle assembly in both neuroblasts and S2 cells [156]. Like its homolog PRC1, Feo binds the KLP3A/KIF4 protein [158], a plus-end directed, microtubule kinesin, required for formation of both the central spindles and the contractile ring [7,153].

The kinesin-6 family member Pavarotti (Pav), the ortholog of human MKLP1, is another conserved protein for bundling central spindle microtubules in *Drosophila* [159]. It has been amply demonstrated that the heterotetramer centralspindlin, composed of two molecules of MKLP1/Pav and two molecules of the Rho family GTPase-activating protein (GAP) MgRacGAP/RacGAP50C (encoded by *tumbleweed*, in *Drosophila*), has a primary role in specifying the site of the cleavage furrow [14,145,160]. In animal cells, the small RhoA GTPase (Rho1 in *Drosophila melanogaster*) is a master regulator of cytokinesis, controlling both F-actin filaments’ formation and contractile ring constriction [145]. The balance between the active (GTP-bound) state and the inactive (GDP-bound) state of RhoA/Rho1 is regulated by the activity of the RhoGEF ECT2/Pebble and the RhoGAP MgRacGAP/RacGAP50C. Research studies in *Drosophila* and mammals have led to the proposal of a model, whereby the RhoGEF ECT2/Pebble interacts with MgRacGAP/RacGAP50C and is targeted to the equatorial cortex by the centralspindlin protein, leading to the local activation of RhoA/Rho1 at the cleavage site [161,162,163,164,165,166]. Consistent with the model, dividing spermatocytes carrying mutations in *pebble* are unable to assemble F-actin rings and fail to undergo cleavage furrow formation [152]. The activities of two serine-threonine kinases, PLK1 and Aurora B, are required to control ECT2/Pebble localization and central spindle formation [13,145]. During anaphase, Aurora B-mediated phosphorylation of MKLP1/Pav controls centralspindlin formation [167,168,169], while the association of RhoGEF ECT2/Pebble with centralspindlin depends on the phosphorylation of MgRacGAP/RacGAP50C by Polo kinase [170,171,172,173].

PLK1 and Aurora B are multifunctional kinases and also control earlier steps of cell division beyond cytokinesis [174,175]. Loss of Polo results in early mitotic arrest in *Drosophila* larval neuroblasts caused by the activation of the spindle checkpoint precluding the analysis of the effects of Polo on cytokinesis [176]. In spermatocytes, which do not have a stringent spindle checkpoint control, a hypomorphic mutant allele in *polo* that does not block meiotic progression disrupts the localization of Pav and the formation of both central spindles and F-actin rings during cytokinesis [177]. During anaphase, Polo accumulates at the spindle midzone and is required for proper CS and F-actin rings formation [177,178]. Consistent with these results, Pav and Polo proteins form a complex and are mutually dependent for localization to the CS midzone in *Drosophila* embryonic cells [159].

Aurora B is the enzymatic subunit of the evolutionarily conserved Chromosomal Passenger Complex (CPC) that contains three additional subunits, Inner Centromere Protein (INCENP), Survivin and Borealin [179]. During mitosis and meiosis, the CPC has a very dynamic localization, which reflects the variety of processes that require its function. Starting from G2, the CPC is associated with chromatin and regulates chromosome condensation; it relocates at inner centromeres from prometaphase until the metaphase-to-anaphase transition and monitors kinetochore–microtubule interaction, correct chromosome alignment and control of the spindle checkpoint [179]. In anaphase, the CPC translocates to the CS microtubules and the equatorial cortex [157,175,180,181,182,183], where it is required for phosphorylation of MKLP1/Pav and CS assembly [167,168,179]. In *Drosophila* spermatocytes, Borealin-related (Borr, the *Drosophila* Borealin) is replaced by its paralog, Australin (Aust) [184]. Aust protein lacks a central region corresponding to a stretch of 140 amino acids of the Borr polypeptide. The reason why spermatocytes need a different component for the CPC is still under investigation; however, it has been shown that the Borr-specific central region is responsible for the interaction with the Shrb subunit of the endosomal sorting complex required for transport-III (ESCRT-III), a molecular complex involved in membrane fission at the end of cytokinesis [185]. It is then likely that in spermatocytes, the replacement of Borr with Aust serves to prevent potentially dangerous associations between the ESCRT-III and the CPC, leading to incomplete cytokinesis. Dividing spermatocytes from males carrying a null *aust* allele exhibit an early cytokinesis arrest with defective CSs and a failure to recruit Pav to the cell equator [184]. Males carrying a hypomorphic allele of *Drosophila INCENP* show similar cytokinesis defects in male meiotic cells, indicating the requirement for the CPC in centralspindlin localization at the cleavage site.

The analysis of a ‘‘separation-of-function’’ allele of *Drosophila* Survivin (dSurvivin encoded by *deterin*) has elucidated the role of Survivin and the CPC, specifically during anaphase and cytokinesis [157]. *Scapolo* (*scpo*) is a *deterin* allele that contains a missense mutation in the baculoviral inhibition of the apoptosis protein repeat (BIR) domain of Survivin. In both spermatocytes and larval neuroblasts, *scpo* allows localization and function of the CPC until anaphase onset, but the CPC and Pav fail to concentrate to the CS and to the equatorial cortex, leading to cytokinesis failure [157]. In spermatocytes, Survivin is also essential to localize Polo and the small Rho1 GTPase to the cell equator. Conversely, in larval neuroblasts from *scapolo* mutants, Polo, Rho1 and Myosin II initially accumulate to a broad equatorial cortical band that fails to convert into a thin contractile ring. The difference between spermatocytes and neuroblasts may be correlated with the existence of a spindle-independent and polarity-dependent furrow-positioning pathway that enables localization of myosin to a broad band in larval neuroblasts during early cytokinesis [186]. However, the spindle-dependent pathway, which depends on Survivin and the CPC, is required to complete cleavage furrow constriction in these cells [186]. In this context, a recent study revealed that peripheral astral microtubules (PAM) and not the interior central spindle microtubules (IMs) play a critical role in dictating and maintaining cleavage furrow positioning during the asymmetric cell division of fly neuroblasts [187]. Thomas and coworkers showed that in these cells, furrowing initiates with the recruitment of a subcortical centralspindlin pool to the basal cleavage site, which depends on PAMs and the CPC. When PAMs are impaired, a distinct midzone-associated pool of centralspindlin becomes dominant, leading to equatorial furrow repositioning and affecting the size asymmetry of the daughter cells [187].

Notably, the analysis of mutants carrying different *polo* alleles showed that the transfer of the CPC to the central spindle in anaphase depends on Polo kinase in male meiosis and neuroblast mitosis [188]. The *polo1* allele contains a point mutation, which causes a substitution of valine 242 by glutamic acid in the kinase domain. The resulting Polo1 mutant kinase is predicted to have reduced enzymatic activity and altered substrate recognition. An analysis of dividing spermatocytes from *polo1* (either homozygous or in combination with other stronger alleles) mutant males revealed that the CPC does not relocate to microtubules during anaphase. Additionally, the defect of CPC localization does not depend on the degree of disruption of the CS, suggesting that the requirement for Polo in cytokinesis could at least in part reflect its role in directing CPC localization. Overall, these data indicate that CS formation and contractile ring assembly require the coordination of the activities of PLK1 and Aurora Kinase, and they depend on each other for a fine tuning of this process.

## 9. Scaffolding Proteins and Phosphoinositide Binding Proteins Contribute to Cleavage Furrow Ingression

During animal cell cytokinesis, the contraction of a contractile ring, composed by non-muscle myosin II (NMII) and F-actin filaments, provides the driving force that generates cleavage furrow ingression [145]. Three different NMII heavy chains (NMHC II) are encoded in humans, whereas just one NMHC II is present in *Drosophila melanogaster*, named Zipper [189,190,191,192]. The NMII hexamer is composed of two identical heavy chains associated with a pair of essential light chains (ELC) and a pair of regulatory light chains (RLC) [193]. The conserved N-terminal globular head domain of the heavy chains has ATPase and actin binding activity. One RLC and one ELC bind with each NMHC II via two IQ motifs on the α-helical neck region, located between the head and the tail domains [193]. The long coiled-coil tail domain at the NMHC II C-terminus enables the assembly of bipolar filaments from NMII monomers [193,194,195,196].

Successful cytokinesis requires that the actomyosin ring is properly anchored to the plasma membrane and connected to the CS [145]. Anillin and the GTP-binding septin proteins are conserved scaffolding proteins at the cleavage site of several organisms, including *Drosophila melanogaster*. Septins are hetero-oligomeric rod-like complexes, which polymerize to assemble non-polar filaments that, in turn, organize higher-ordered structures [197,198,199,200,201]. In *Drosophila melanogaster*, five septins have been identified [202]. Three septins, namely Peanut (mammalian SEPT7 ortholog), Sep1 (mammalian SEPT2) and Sep2 (mammalian SEPT1), have been localized to contractile rings and ring canals in *Drosophila* male germ cells [7,203,204]. Septins bind phosphoinositides in vitro [200,201,205,206], which promotes septin polymerization [207]. A pivotal role as a scaffolding protein in the contractile ring of dividing spermatocytes is played by the evolutionarily conserved protein anillin. Originally identified as an F-actin binding protein from *Drosophila* embryo extracts, anillin is a conserved protein that bundles F-actin filaments [208,209,210] and also associates with NMII and septins, which are also filament-forming proteins [202,211,212]. Anillin contains multiple conserved protein domains that enable its interaction with several proteins required for cytokinesis [213,214,215], which include RhoGTP1 [216,217], RacGAP50C [218,219], Citron kinase/Sticky [220] and ECT2/Pebble [221]. Moreover, anillin binds membrane phosphoinositides [222]. Based on these data, anillin is an ideal candidate for mediating interactions between the CS microtubules and the equatorial cortex to stabilize the contractile ring apparatus. In dividing spermatocytes, anillin starts to accumulate to the cleavage site of dividing spermatocytes in anaphase before the appearance of F-actin [151]. Consistent with this temporal order, spermatocytes from mutants in the gene *chickadee* (*chic*), which encodes the actin binding protein profilin, display initial accumulation of anillin but fail to assemble an F-actin ring at the cell equator. Goldbach and coworkers analyzed the effects of mutations in anillin on spermatocyte cytokinesis. They reported that anillin is required to recruit the septins Peanut and Sep2 to the cleavage furrow and to maintain Rho1, F-actin and Myosin II during late cytokinesis. By using FRAP experiments, they further demonstrated that anillin, septins and the RLC Spaghetti Squash (Sqh) are stably associated with the cleavage furrow and do not undergo rapid exchange during contractile ring constriction in wild-type spermatocytes. Conversely, FRAP experiments on anillin-depleted spermatocytes revealed that Sqh completely loses its association with the cleavage furrow.

Cytokinesis also implicates the addition of new membrane achieved through the delivery of membrane vesicles to the cleavage furrow and involving both the secretory and endocytic/recycling pathways [145,223]. *Drosophila* male meiosis has proved to be an excellent cell system to identify vesicle trafficking proteins required for cytokinesis [14]. Screens for mutants defective in spermatocyte cytokinesis have identified Golgi proteins as well as proteins of endocytic/recycling machinery, including the conserved oligomeric Golgi complex (COG) subunits Cog5 and Cog7; the TRAPPII complex subunit Brunelleschi; the syntaxin 5 ER-to-Golgi vesicle-docking protein; the small GTPases Rab11, Rab1 and Arf6; the COPI subunits and the exocyst complex proteins Sec8 and Exo84 [224,225,226,227,228,229,230,231,232,233].

Recent data have pointed out that phosphoinositides in the cleavage furrow plasma membrane, particularly phosphatidylinositol 4,5-bisphosphate [PI(4,5)P2], are crucial molecules for the assembly and maintenance of the cytokinesis apparatus and for coordinating contractile ring dynamics with membrane trafficking [204,209,234,235,236]. The phosphoinositide PI(4,5)P2 not only binds anillin and septins and influences their localization [207,222], but it also regulates F-actin filament assembly by modulating the activity of the actin binding proteins profilin, which is also required for spermatocyte cytokinesis [7,237]. The centralspindlin subunit CYK-4/RacGAP50C/MgcRacGAP (RacGAP50C) also associates with the furrow plasma membrane by binding to phosphatidylinositol 4-phosphate [PI(4)P] and PI(4,5)P2 phosphoinositides via its C1 domain [238]. *Drosophila* PI(4)P-binding protein Golgi phosphoprotein 3 (GOLPH3, also known as Sauron) accumulates at the cleavage site of dividing spermatocytes and is required to maintain centralspindlin at the cell equator and to stabilize myosin II and septin rings [204]. The loss of GOLPH3 disrupts the localization of PI(4)P-enriched secretory organelles at the cleavage furrow. Reciprocally, GOLPH3 function in cytokinesis depends on its PI(4)P binding pocket, suggesting that this protein has a key function in coupling PI(4)P signaling, membrane trafficking and actomyosin ring formation [204,239,240,241].

The molecular mechanisms that concentrate NMII filaments at the equatorial cortex of dividing cells are not completely understood. The dynamics of NMII filaments have been studied by total internal reflection fluorescence microscopy in flattened dividing *Drosophila* S2 cells [242]. This study revealed that NMII concentrates at the equatorial cortex and disappears from the poles during anaphase [242]. The small GTPase RhoA (Rho1 in *Drosophila*) is known to control NMII by activating Rho kinase, which, in turn, phosphorylates RLC on Thr-18 and Ser-19 [243,244,245,246]. It has been proposed that NMII filaments concentrate at the equatorial cortex through a mechanism that requires Rho kinase-mediated RLC phosphorylation. However, in contrast with this model, imaging of truncated NMHC II proteins in dividing *Drosophila* S2 cells revealed that myosin filaments concentrate to the equatorial in the absence of Rho1 [247]. The characterization of a male sterile mutant carrying on a missense allele of the *Drosophila* gene *zipper* indicates that interaction with plasma membrane lipids plays an important role in localizing NMII at the cleavage furrow [240]. *Celibe* (*cbe*) mutants carry a mutation affecting the conserved region of the Myosin heavy chain protein Zipper (Zip), replacing a conserved isoleucine residue adjacent to the RLC-binding IQ motif. *cbe* impairs binding of the Zip protein to the RLC Sqh. Dividing spermatocytes from *cbe* mutant males show initial localization of Zip and septin proteins at the cell equator. However, Sqh fails to concentrate, resulting in thin contractile rings that fragment during later stages of cytokinesis. These results are consistent with the data from Beach and Egelhoff showing that mammalian NMHC II localize to the cleavage furrow independently of RLC [248] and indicating that the mechanisms that target NMHC II at the equatorial cortex do not depend on RLC (Sqh in *Drosophila*) phosphorylation. However, Zip–Sqh interaction is required for the formation of robust actomyosin rings and contractile ring constriction.

Because the hydrophobic amino acid isoleucine is often involved in binding to hydrophobic ligands, including lipids, it is likely that its replacement with the aromatic amino acid phenylalanine in *cbe* might affect the interaction of NMHC II with the plasma membrane and the assembly of myosin thick filaments at the cleavage site. Consistent with this hypothesis, NMII proteins bind to liposomes containing one or more acidic phospholipids, and the binding of NMIIs to negatively charged liposomes occurs predominantly through the RLC-binding site of the heavy chain [249]. Collectively, these results suggest that membrane-bound monomers can bind to RLC and initiate polymerization of myosin filaments. The phosphoinositide PI(4,5)P2, which is highly enriched at the cleavage site of dividing spermatocytes [204,234], is the likely candidate to mediate the interaction between Zip and the furrow plasma membrane. *cbe* mutation also disrupts localization of GOLPH3 protein and maintenance of centralspindlin at the cell equator of telophase cells. GOLPH3 protein associates with Sqh and directly binds the C-terminal domains of the kinesin-like protein Pav, suggesting that it might control the stability of centralspindlin at the invaginating plasma membrane. Overall, the findings in this study propose that the reciprocal dependence between NMII filaments and PI(4)P–GOLPH3 regulate centralspindlin stabilization and contractile ring function during spermatocyte cytokinesis (Figure 3).

## 10. Interconnectivity during *Drosophila* Spermatogenesis

As described above, cytokinesis during the mitotic and meiotic cell divisions of male germ cells is incomplete, resulting in intercellular bridges called ring canals (RCs) of 1–2 microns in diameter, that connect germ cells in each cyst (Figure 4) [250,251].

RCs were originally discovered in ovarian germ cells, where they play an important role for oocyte growth during oogenesis [250,252]. In *Drosophila* males, RCs are maintained during spermatid elongation and individualization [9,203]. It is important to note that after the completion of the four germ cell divisions that precede meiosis, the biogenesis and the structure of male RCs diverge significantly from female RCs [253,254]. The RCs of male germ cells contain components of the arrested cleavage furrows, including anillin and the septins Sep1, Sep2 and Peanut (Pnut), but lack some of the proteins that stabilize the actin-rich RCs of the female germline, such as Kelch and hu-li tai shao-RC [9,151,203,250]. Other molecular components of the male RCs include the kinesin-like protein Pav and its partner RacGAP50C, the cytoskeletal scaffold protein Cindr, the centralspindlin partner Nessun Dorma (Nesd) and the microtubule-associated protein Orbit/CLASP [9,154,203,251,253,255,256,257]. The functions of RCs that connect male germ cells have been less characterized, although it has been suggested that male RCs might allow the sharing of cytoplasmic signals and the promotion of cell synchronization (e.g. meiotic entry) [254]. Using a combination of live cell imaging and genetics, Kaufman and coworkers demonstrated intercellular trafficking of GFP and endogenous proteins through the RCs of male germ cell cysts in all stages of spermatogenesis [9]. RCs allow intercellular movement of some proteins within the germline cysts. However, protein size does not correlate with the protein’s ability to move into neighboring cells, and there are proteins (for example, calmodulin) that can move freely through ovarian follicle cell RCs but not through testis RCs [9], suggesting the existence of an active but still uncharacterized control mechanism. The reason of this movement after male meiosis is not completely clear, although it has been suggested that RCs might drive X-encoded products to Y-bearing spermatids [9,258,259] and vice versa to X-bearing spermatids. However, the need of a shared cytoplasm in pre-meiotic phases is still a debated topic. In this context, a recent work suggested that interconnectivity may help cells to synchronize a triggered response, i.e., cell death, in cases of stress, such as after DNA damage caused by ionizing radiation [260], thus providing a concerted response that eliminates a damaged cyst in its entirety, irrespective of the damage endured by individual cells.

In addition to RCs, developing germ cells within a cyst are connected by the fusome, a branched organelle that extends through each RC (Figure 4). The fusome contains several cytoskeleton proteins, the most abundant being α- and β-Spectrin (encoded by the genes α-*spec* and β-*spec*, respectively), Ankyrin (encoded by the *ank* gene) and Adducin (encoded by the gene *hu li tai shao*, *hts*) [261,262,263]. Different from female fusomes that connect only pre-meiotic cells, male fusomes are present throughout spermatogenesis [203] and do not contain endoplasmic reticulum (ER) proteins [8,9,251]. To better understand the role of the fusome in *Drosophila* male germ cells, Kaufman and coworkers performed RNAi experiments targeting the fusome structural proteins α-Spectrin and Adducin [9]. In both cases, they observed the complete disappearance of the fusome or its fragmentation, with only a small subset of fusome fragments associated with RCs. Despite the extensive fragmentation of the fusome, the germ cell cysts were still able to localize Pav to RC, and the movement of molecules through RC was only slightly affected (being faster in the absence of an intact fusome). The diameter of RCs was not significantly altered, although a subset of them (11% ca.) showed collapsed or non-circular RC lumen, suggesting a role of the fusome in RC morphology and/or stability [9]. Moreover, fusome disruption caused minimal effects on male fertility, in contrast with data obtained on female fusomes, where its disruption causes complete sterility [262,264]. It is possible that the fusome plays a role in the selection of what can pass through RCs, for example, allowing the passage of cell division signals but blocking the movement of organelles, such as mitochondria or centrosomes. Consistent with this hypothesis, *hts* mutants show centrosome movements through the RC, which does not occur in wild-type males, thus altering their number inside the cells [265]. Nonetheless, the role of the fusome in male germline and its significance for male fertility require additional studies to be fully elucidated.

## 11. Conclusions

The comparison of *Drosophila* and human genomes revealed that more than 75% of the genes mutated in human diseases have a *Drosophila* counterpart, which is conserved not only in its gene sequence, but also in its function and (at least some) phenotypic characteristics [266]. For this reason, *Drosophila melanogaster* is currently and extensively exploited for biological research to study the molecular mechanisms underlying many human diseases, such as neurodegenerative and metabolic disorders, cardiovascular diseases, cancer and infertility [267,268,269].

Centrosome amplifications, arising from defects in centrosome duplication or cytokinesis failures, represent a hallmark of human cancer and have been implicated in the initiation and progression of cancer [270]. Thanks to the sophisticated genetic tools and the unique cellular characteristics, *Drosophila* male meiosis has proven to be a powerful model system for studying the molecular mechanisms that regulate centriole duplication and centrosome assembly, spindle assembly and cytokinesis in the context of a multicellular organism. Owing to the large size of these cells, the centrosome-mediated mechanisms of spindle assembly and the large size of the centrioles, *Drosophila* spermatocytes have significantly contributed to our knowledge of centrosome architecture and cycle [45].

Forward genetic or RNA interference-based screens have contributed to identify many molecular players of *Drosophila* spermatocyte cytokinesis [13,152]. Evidence indicates that perturbations of cytokinesis result in stronger phenotypes in dividing spermatocytes when compared with dividing S2 cells and larval neuroblasts, indicating that male meiosis is more sensitive to mutations affecting cytokinesis than somatic cells [13]. Importantly, spermatocytes are much more dependent on membrane trafficking pathways, making them an ideal cell system to investigate how membrane remodeling is coupled with central spindle and contractile ring dynamics during cleavage furrow ingression.

Cytokinesis failures lead to tetraploidy, which preludes chromosome instability and has been correlated with tumor progression, metastasis and with multi-drug resistance [271,272]. Several proteins involved in spermatocyte cytokinesis have been implicated in cancer pathogenesis, such as GOLPH3, myosin II and anillin [273,274,275,276]. Thus, investigating the molecular pathways that require these proteins in cytokinesis opens up new avenues for cancer diagnosis and treatment. In this context, an approach based on affinity purification coupled with mass spectrometry (AP-MS) has allowed identifying of the protein–protein interaction network of *Drosophila* GOLPH3 (dGOLPH3) in testes. Remarkably, the dGOLPH3 interactome is enriched with proteins that control cell cycle progression and cell proliferation, suggesting new molecular targets that might be relevant for therapeutic interventions in cancer [241].

## Figures and Tables

**Figure 1 cells-11-00695-f001:**
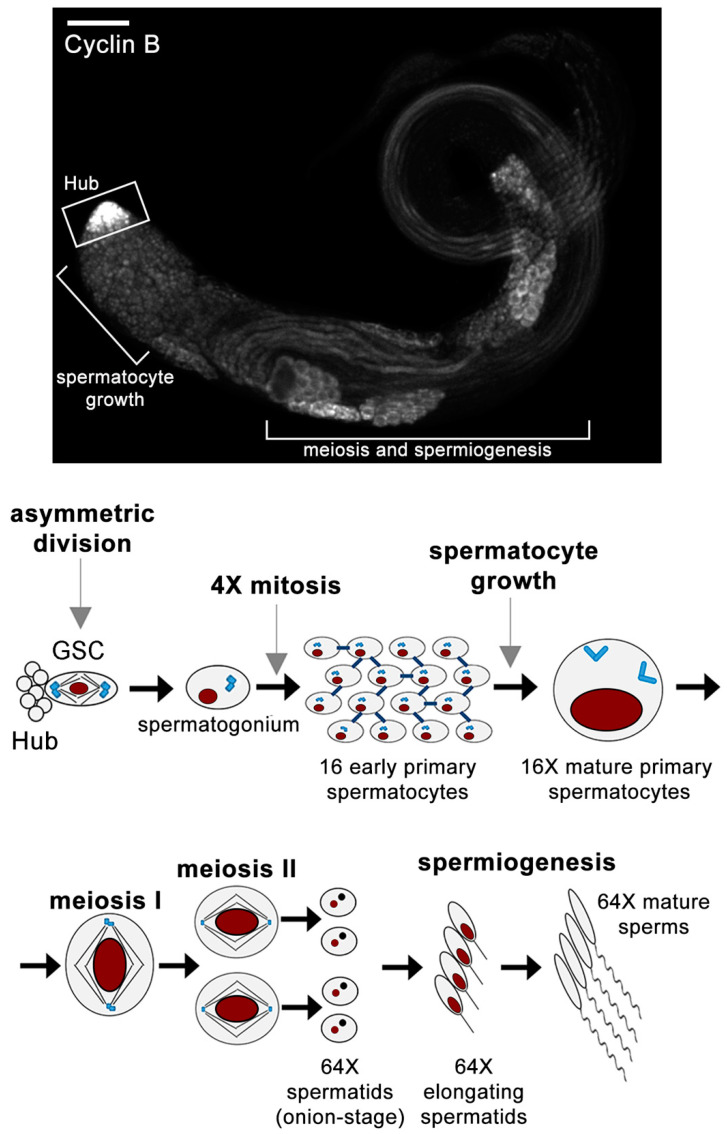
Spermatogenesis in *Drosophila melanogaster*. Schematic showing the different stages of spermatogenesis from asymmetric cell divisions of germline stem cells (GSCs) to the mature sperm. The 16 SGs in each cyst progress to the premeiotic S phase and then embark on the G2 phase and become spermatocytes. During the G2 phase, the cell volume grows 25-times, while spermatocytes undergo robust gene expression, and the centrioles dock at the plasma membrane, where they assemble short cilia on all four centrioles (light blue). Cyclin B1 (Cyclin B) protein accumulates in the cytoplasm of the spermatocyte during the late G2 phase (upper panel). At the onset of meiosis, the centriole pairs (together with their short cilia) dislodge from the plasma membrane, move back towards the nucleus and nucleate astral microtubules. Mature spermatocytes undergo two meiotic divisions in rapid succession, with incomplete cytokinesis, resulting in 64 haploid round spermatids connected by RCs. Following meiosis, all the mitochondria contained in each spermatid fuse to form a complex interlaced structure named the nebenkern. Cysts of wild-type early round spermatids contain a total of 64 interconnected spermatids, each containing a single phase-dark nebenkern (black) associated with a single phase-light nucleus (dark purple) of a similar size. Chromosomes and nuclei have the shape of an ellipse (dark purple). Scale bar, 10 µm.

**Figure 2 cells-11-00695-f002:**
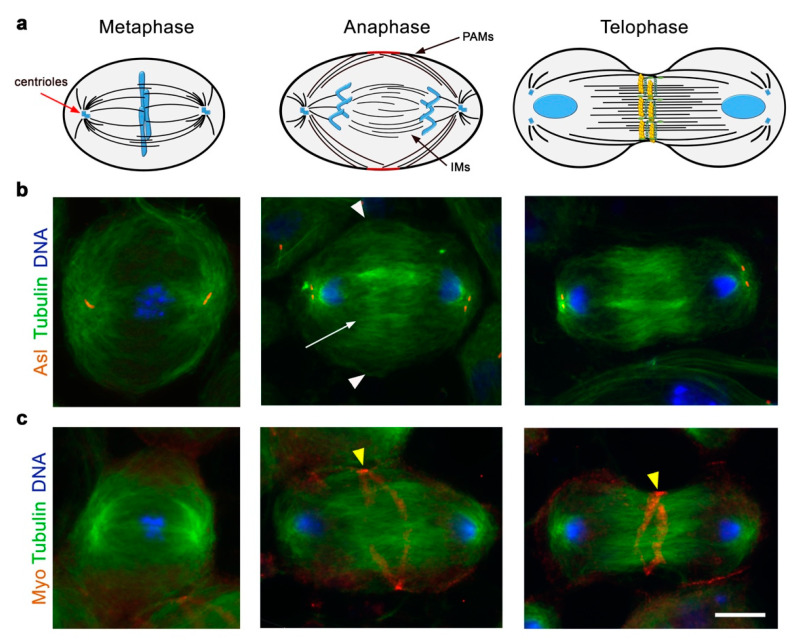
Dividing spermatocytes as a model system to investigate the dynamics of central spindle microtubules and actomyosin rings during cytokinesis. (**a**) The schematic (upper panel) depicts different stages of male meiotic division. Central spindle microtubules direct the position of the cleavage site (red) at the equatorial cortex in a position that bisects the axis of chromosome segregation. Late anaphase and telophase spermatocytes exhibit a prominent central spindle, an array of antiparallel interdigitating microtubule bundles between segregated chromosomes. During late anaphase, the central spindle is made up of two distinct microtubule populations: the peripheral astral microtubules (PAMs) and the interior central spindle microtubules (IMs). The contractile ring, composed of F-actin filaments and non-muscle Myosin II, is assembled on the inner face of the plasma membrane around the cell equator of late anaphase spermatocytes. Microtubules are depicted in black, the actomyosin ring in yellow and green and chromosomes and nuclei in light blue. (**b**) Primary spermatocytes, which are stained for tubulin, the centriolar protein Asl and DNA, are during metaphase, late anaphase and early telophase. White arrowheads point to PAMs, and the arrow indicates IMs. (**c**) Primary spermatocytes, which are stained for tubulin, the contractile ring protein Zipper (Myo) and DNA, are during metaphase, late anaphase and early telophase. Dividing spermatocytes assemble Zip-contractile rings at the equatorial cortex during late anaphase–early telophase. Yellow arrowheads indicate the contractile rings. Scale bar, 10 μm.

**Figure 3 cells-11-00695-f003:**
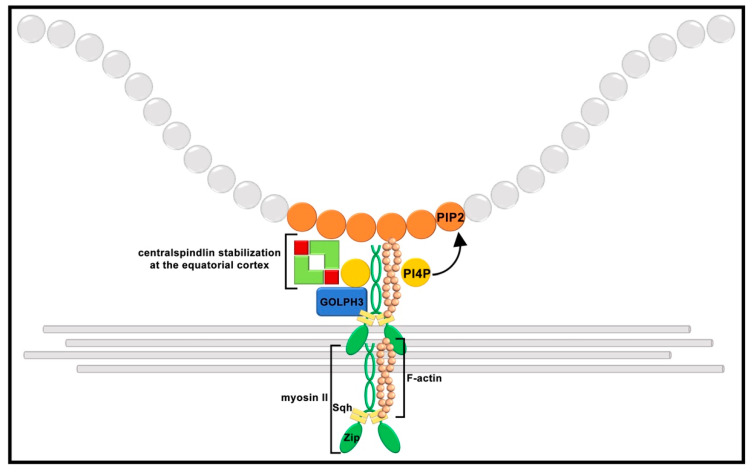
The reciprocal dependence of non-muscle myosin II and PI(4)P–GOLPH3 regulates centralspindlin stabilization at the invaginating membrane. The binding of Zip protein to Sqh is required for robust actomyosin ring formation at the cleavage furrow. The Zip–Sqh complex interacts with GOLPH3 protein and controls localization of this protein and PI(4)P (PI4P) to the cleavage furrow. In turn, GOLPH3 binds the kinesin-like protein Pav and is required for centralspindlin maintenance at the invaginating plasma membrane. The interaction of RacGAP50C with PI(4)P and PI(4,5)P2 (PIP2) also contributes to centralspindlin stabilization at the furrow plasma membrane.

**Figure 4 cells-11-00695-f004:**
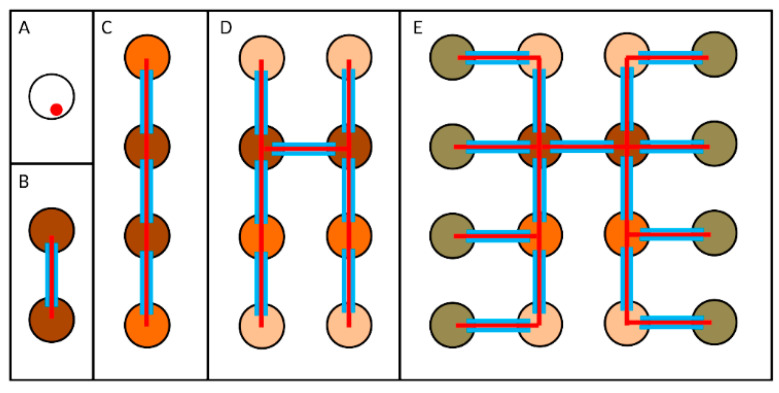
Interconnectivity during *Drosophila* male germ cells. A schematic showing the four rounds of mitotic divisions (**A**–**E**) to produce a cyst of 16 germ cells, which undergo S phase and become spermatocytes (panel **E**). In each panel, cells (circles) originated during the same division are indicated with the same color, and different colors are used for each division. The red color indicates the fusome (**B**–**E**), while blue lines represent the ring canals (RCs); the size of RCs and fusomes is exaggerated for figure readability.

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
