# Peer review of "Microtubule and Actin Cytoskeletal Dynamics in Male Meiotic Cells of Drosophila melanogaster"

_cells, 2022, doi:10.3390/cells11040695_

Round 1
Reviewer 1 Report
This manuscript by Frappaolo et al. summarizes knowledge on Drosophila spermatogenesis with a focus on meiosis and underlying regulation of the cytoskeleton. With 278 references, the text of the review is informative, well written, extremely complete and up-to-date. I have the following comments:
1) On figures:
In Fig 1b, Asterless and Anillin are represented with the same color. Please, separate them.
Please, define acronyms again in figure legend to increase readability.
The authors should consider adding one or more figures to the review to increase its usefulness. Compared to the sophisticated understanding we possess of this system, the complexity of the findings described and the amount of work referenced in the review, the two figures feel a bit too simple. Actin is not even represented. I would suggest at least a figure on the broad topic of cytoskeleton regulation, the leitmotiv of the review. Perhaps also another one on the the interaction with cytoskeleton of Golgi protein Golph3, on which the Giansanti lab has contributed major findings.
2) There are many instances of unnecessary commas. Please, revise the whole text. A few examples:
L49 all the mitochondria, contained in each spermatid, fuse
L61 ring canals (RCs), that interconnect
L105 Testes from mutants in twine/Cdc25, undergo spermatid differentiation, despite failure of the spermatocytes to undergo the meiotic division
L110 stage-specific repression of translation, restricts CycB protein accumulation
L120: germ cells from eIF4G2 mutant testes, skip the major events of meiotic division
3) Regarding section 2, on translational control of cytoskeleton gene expression:
It is a very long read from L82 to L95 until one finds out what all this has to do with cytoskeleton. Please, focus the topic earlier.
L103-112 Please, make connection to cytoskeleton explicit.
In general, I think section 2 should be rewritten in a way that highlights and brings to the front specific instances where translational control of cytoskeleton gene expression is understood.
4) L391 due TO the lack
5) L468 In spermatocytes, WHICH do not have
Author Response
Referee #1
This manuscript by Frappaolo et al. summarizes knowledge on Drosophila spermatogenesis with a focus on meiosis and underlying regulation of the cytoskeleton. With 278 references, the text of the review is informative, well written, extremely complete and up-to-date. I have the following comments:
1) On figures:
In Fig 1b, Asterless and Anillin are represented with the same color. Please, separate them.
Please, define acronyms again in figure legend to increase readability.
We have modified Figure 1 and removed panel b.
The authors should consider adding one or more figures to the review to increase its usefulness. Compared to the sophisticated understanding we possess of this system, the complexity of the findings described and the amount of work referenced in the review, the two figures feel a bit too simple. Actin is not even represented. I would suggest at least a figure on the broad topic of cytoskeleton regulation, the leitmotiv of the review. Perhaps also another one on the interaction with cytoskeleton of Golgi protein Golph3, on which the Giansanti lab has contributed major findings.
Following the suggestions, we have one Figure (new Figure 2) to illustrate the dynamics of central spindle microtubules and the actomyosin cytoskeleton in dividing spermatocytes
In addition, we have added a new figure (new Figure 3) to illustrate the reciprocal dependence of non-muscle myosin II and PI(4)P-GOLPH3 in centralspindlin stabilization at the invaginating membrane during spermatocyte cytokinesis.
2) There are many instances of unnecessary commas. Please, revise the whole text. A few examples:
L49 all the mitochondria, contained in each spermatid, fuse
L61 ring canals (RCs), that interconnect
L105 Testes from mutants in twine/Cdc25, undergo spermatid differentiation, despite failure of the spermatocytes to undergo the meiotic division
L110 stage-specific repression of translation, restricts CycB protein accumulation
L120: germ cells from eIF4G2 mutant testes, skip the major events of meiotic division
We have deleted the unnecessary commas
3) Regarding section 2, on translational control of cytoskeleton gene expression:
It is a very long read from L82 to L95 until one finds out what all this has to do with cytoskeleton. Please, focus the topic earlier.
We have modified section 2 to focus the topic earlier, see pages 3,4, lines 87-281
L103-112 Please, make connection to cytoskeleton explicit.
In general, I think section 2 should be rewritten in a way that highlights and brings to the front specific instances where translational control of cytoskeleton gene expression is understood.
We have modified former L103-112 to make connection to cytoskeleton remodeling more explicit. See L88-L97 of the revised MS.
4) L391 due TO the lack
We have corrected the typo.
5) L468 In spermatocytes, WHICH do not have
We have corrected the typo.

Reviewer 2 Report
This manuscript by Anna Frappaolo with coauthors named ‘Microtubule and actin cytoskeletal dynamics in male meiotic cells of Drosophila melanogaster contains a very detailed review devoting a very interesting theme. Actually, Drosophila spermatocytes due to their large size, sophisticated genetic tools, and other unique cellular characteristics provide an excellent model system for uncovering the mechanistic basis and molecular pathways regulating spindle microtubules dynamics, remodeling of actin cytoskeleton systems, and cytokinesis process through meiosis. Based on the knowledge accumulated over several decades, the authors provide an extensive and detailed bibliographical review of their favorite working experimental model. It seems the manuscript appropriately covers the significant literature, spanning a very long period from the early sixties-eighties to articles that have been published very recently.
In whole, before I can recommend that this review be accepted, firstly I propose that some additional improvements should be performed in the language quality and style. There are two main characteristics of this manuscript that severe obscure the understanding of the review:
- To aim to describe the molecular processes associated with the dynamic cytoskeletal organization of meiotic spermatocytes, the authors included rather excess data descriptions in their review. Often this is descriptive and it is hard to follow logic. To my mind, it would be useful to improve the whole structure of the review and to make summarizing statements at the end of each section to facilitate the perception of the data.
- While the authors included two figures in the review, these images are rather confusing. The first of them presented a general scheme of spermatogenesis, but it is not associated with the main theme of the review. It should also be noted that this spermatogenesis scheme is very small and incomprehensible. I recommend that this scheme be adjusted. Figure 2 is devoted to interconnectivity between Drosophila male germ cells. Without questioning the usefulness of this figure for understanding the relevant part of the review, I would like to say that many from the other seven sections need more understandable schemes or tables.
In sum, although the manuscript describes a large amount of data on the topic, it is very difficult for readers to understand it without the necessary remodeling that provides the logic of presenting the material, analysis, and conclusions in each section, as well as without adequate illustrative material.
Author Response
Referee#2
- To aim to describe the molecular processes associated with the dynamic cytoskeletal organization of meiotic spermatocytes, the authors included rather excess data descriptions in their review. Often this is descriptive and it is hard to follow logic. To my mind, it would be useful to improve the whole structure of the review and to make summarizing statements at the end of each section to facilitate the perception of the data.
We have re-ordered the text and removed the excess data descriptions in sections 2 and section 6 (former section 5). We divided section 3 into 2 paragraphs, each to provide greater clarity. Moreover, we added summarizing statements at the end of section 4 to facilitate the perception of the data (page 6, L753-755).
- While the authors included two figures in the review, these images are rather confusing. The first of them presented a general scheme of spermatogenesis, but it is not associated with the main theme of the review. It should also be noted that this spermatogenesis scheme is very small and incomprehensible. I recommend that this scheme be adjusted.
The general scheme of spermatogenesis has been enlarged and included in new Figure 1.
Figure 2 is devoted to interconnectivity between Drosophila male germ cells. Without questioning the usefulness of this figure for understanding the relevant part of the review, I would like to say that many from the other seven sections need more understandable schemes or tables.
We have now included two new Figures. New Figure 2 describes the dynamics of central spindle microtubules and actomyosin rings during cytokinesis.
New Figure 3 describes the reciprocal dependence of non-muscle myosin II and PI(4)P-GOLPH3 in centralspindlin stabilization at the invaginating membrane during spermatocyte cytokinesis.
In sum, although the manuscript describes a large amount of data on the topic, it is very difficult for readers to understand it without the necessary remodeling that provides the logic of presenting the material, analysis, and conclusions in each section, as well as without adequate illustrative material.
We have provided new illustrative materials (two new Figures) and modified the text throughout the manuscript. Moreover, we removed the excess of data description (sections 2 and 6) and divided one paragraph into two specific sections. We hope that the manuscript is now more readable.

Reviewer 3 Report
The paper of Frappaolo et al. entitled “Microtubule and actin cytoskeletal dynamics in male meiotic cells of Drosophila melanogaster” is a timely review on the role of microtubules and microfilaments during the spermatogenesis of Drosophila melanogaster. The AA analyse in detail the mechanisms involved in the process of cell division during male meiosis and show how the molecular pathway of this process is highly conserved. This review is well written and interesting and can contribute to a better understanding of the relationship between some cytoskeletal elements and the process of meiotic division.
It would be great if Authors can address the role of the centrosome during the spermatogonial divisions and during the meiotic divisions. Anyway, in my opinion the manuscript could be published in the present form.
Author Response
Referee #3
It would be great if Authors can address the role of the centrosome during the spermatogonial divisions and during the meiotic divisions. Anyway, in my opinion the manuscript could be published in the present form.
We have now added a section to illustrate the different role of centrosomes during spermatogonial divisions and during meiotic divisions. See L685-694, page 6.

Round 2
Reviewer 2 Report
I appreciate some re-ordering of the revised text of review provided by authors and the inclusion of two additional illustrations. Yes, to my limited knowledge of this field, this review seems to deeply cover the significant literature data, spanning a very long period until today. Given the big work done, I agree that this review can be accepted and be useful for biologists working in the relevant field.
Author Response
Thank you for your comments. We appreciate that the review is now acceptable.